# PM_2.5_ and Trace Elements in Underground Shopping Districts in the Seoul Metropolitan Area, Korea

**DOI:** 10.3390/ijerph18010297

**Published:** 2021-01-03

**Authors:** Soo Ran Won, In-Keun Shim, Jeonghoon Kim, Hyun Ah Ji, Yumi Lee, Jongchun Lee, Young Sung Ghim

**Affiliations:** 1Indoor Environment and Noise Research Division, National Institute of Environmental Research, Incheon 22689, Korea; wsr1984@korea.kr (S.R.W.); inkeun77@korea.kr (I.-K.S.); kimxls88@korea.kr (J.K.); saya006@korea.kr (H.A.J.); dbal1808@korea.kr (Y.L.); roundvoid@korea.kr (J.L.); 2Department of Environmental Science, Hankuk University of Foreign Studies, Yongin, Gyeonggi 17035, Korea

**Keywords:** indoor air quality, trace element sources, outdoor influence, anthropogenic influence, cancer risk

## Abstract

We measured PM_2.5_ in 41 underground shopping districts (USDs) in the Seoul metropolitan area from June to November 2017, and associated 18 trace elements to determine the sources and assess the respiratory risks. The PM_2.5_ concentrations were 18.0 ± 8.0 μg/m^3^ inside USDs, which were lower than 25.2 ± 10.6 μg/m^3^ outside. We identified five sources such as indoor miscellanea, soil dust, vehicle exhaust/cooking, coal combustion, and road/subway dust, using factor analysis. Almost 67% of the total trace element concentration resulted from soil dust. Soil dust contribution increased with the number of stores because of fugitive dust emissions due to an increase in passers-by. Vehicle exhaust/cooking contribution was higher when the entrances of the USDs were closed, whereas coal combustion contribution was higher when the entrances of the USDs were open. Although miscellanea and coal combustion contributions were 3.4% and 0.7%, respectively, among five elements with cancer risk, Cr and Ni were included in miscellanea, and Pb, Cd, and As were included in coal combustion. The excess cancer risk (ECR) was the highest at 67 × 10^−6^ for Cr, and the ECR for Pb was lower than 10^−6^, a goal of the United States Environmental Protection Agency for hazardous air pollutants.

## 1. Introduction

PM_2.5_ refers to particular matter (PM) having a diameter less than 2.5 μm. PM_2.5_ introduced in the human body through breathing may cause respiratory and cardiovascular diseases and even premature death by infiltrating the lungs, alveoli, and blood vessels [1]. Every 10 μg/m^3^ increase in short-term exposure to PM_2.5_ concentrations caused a 0.38% increase in mortality owing to respiratory and cardiovascular diseases [2,3]. In addition, PM_2.5_ acts as a medium that introduces toxic substances in the human body because it facilitates the binding of polycyclic aromatic hydrocarbons, trace elements, bacteria, and viruses to large surface areas with high absorptivity [4]. Water-soluble metals in PM are reported to increase cancer risks by causing oxidant damage to the DNA of human airway epithelial cells [5].

People living in cities spend approximately 80–90% of their time indoors in houses, schools, public transportation, and shopping malls, which increases their exposure to pollutants emitted from construction materials, home appliances, and electronic products [6,7]. The construction of subway stations and large buildings due to the development of large-scale downtown areas has boosted the need for efficient space utilization in Korea. Underground shopping districts (USDs) have been popular in the country since the 1970s; they are used for pedestrian traffic and as evacuation facilities. USDs are “public-use facilities” managed by the Ministry of Environment (MOE) since 1996 under the Indoor Air Quality Control Act [8]. The term “public-use facilities” refers to facilities used by the general public. Indoor air quality is currently managed for 10 pollutants: PM_10_, PM_2.5_, CO_2_, formaldehyde, total airborne bacteria, CO, NO_2_, radon, total volatile organic compounds (TVOC), and mold. As per the Korean indoor air quality standards for USDs, the 24 h average PM_10_ concentration should not exceed 100 μg/m^3^ (until 1999, this value was 250 μg/m^3^). Since 2018, the corresponding standard for PM_2.5_ is 50 μg/m^3^. Notably, different standards exist for indoor and outdoor air. The 24 h standards for outdoor air are 100 μg/m^3^ for PM_10_ and 35 μg/m^3^ for PM_2.5_ [9]; the PM_2.5_ standard was tightened from 50 μg/m^3^ in March 2018, reflecting public concern. The government has been making considerable efforts to manage PM pollution because it is designated as a Group 1 carcinogen by the International Agency for Research on Cancer, and the public is significantly affected by high PM concentrations.

Studies on indoor air pollution have typically focused on gaseous pollutants (e.g., VOCs, formaldehyde, and CO_2_ in department stores and large shopping malls) [10,11,12,13,14] and the sick building syndrome [15,16,17,18]. Research on indoor pollution sources, such as cooking and smoking, has also been consistently conducted, but the number of such studies is limited [19,20,21,22,23,24,25]. According to Karagulian et al. [26], approximately 200 studies are related to the estimation of atmospheric pollution sources, but few focus on indoor air pollution [27,28], reporting on the effects of indoor and outdoor sources for residences and schools. In addition, the health risks of trace elements have been investigated consistently since the 2000 s [29,30,31,32]. Despite very low concentrations, some elements such as As, Cd, Cr, and Pb pose high health risks [33,34,35]. However, research related to indoor sources of trace elements and their health risks is insufficient. Thus, in this study, we measured PM_2.5_ and associated trace elements in USDs in the Seoul metropolitan area (SMA), Korea (Figure 1), whose indoor air quality is managed by the government. We estimated sources of trace elements using factor analysis, and examined the effects of the outdoor emissions (vs. the generation within USDs) and anthropogenic emissions (vs. crustal origin). Finally, we attempted to assess the excess cancer risk (ECR) caused by respiratory exposure to selected elements in USDs.

## 2. Experimental Method

### 2.1. Study Sites

The SMA in the northwestern part of Korea includes Seoul proper, Incheon, and Gyeonggi (Figure 1). According to the Korean Statistical Information Service, as of 2018, the populations of Seoul, Incheon, and Gyeonggi were 9,673,936, 2,936,117, and 13,103,188, respectively, with a total country population of 51,629,512 [36]. Thus, approximately 50% of the total population was concentrated in SMA. Accordingly, a larger number of public-use facilities are also located in SMA. Among 64 USDs managed by the MOE, 42 (25 in Seoul, 15 in Incheon, and 2 in Gyeonggi) are located in SMA. The area of each USD ranges from 777 to 31,566 m^2^, with the corresponding years of construction spanning from 1967 to 2009. Depending on the facility size, there are 15 to 624 stores in each USD. We divided USDs into “open,” “semi-open,” and “closed” types, based on the style of entrances. The majority of the stores (approximately 76%) sold sundry goods such as clothes, shoes, or bags, followed by cellphone stores (6%), cosmetics (5%), snack bars (3%), and nail shops (3%). We also divided USDs into “open,” “mixed,” and “closed,” depending on the entrance type between the stores and the passageways. For closed-type facilities, stores are separated from the passageway, whereas for the open-type facilities, stores are connected directly to a passageway and have no entrance. Mixed-type facilities have stores with and without entrances to the passageways. Of the 42 USDs, 23 were connected to subway underground stations.

### 2.2. PM_2.5_ Sampling

As the entrance, store type, areas, and number of passers-by differ considerably across the USDs, we determined the number of sampling points according to the standard methods for indoor air quality specified by the National Institute of Environmental Research, Korea [37]. The number of sampling points was two for an area smaller than 10,000 m^2^, three for an area between 10,000 and 20,000 m^2^, and four for an area larger than 20,000 m^2^. We sampled outdoor air at one point near the entrance of each USD. The sampling period spanned from June to November 2017, and sampling was performed on three consecutive days for each USD. One of the 42 USDs was excluded from sampling due to extensive renovations.

We performed sampling for 24 h at a flow rate of 5 L/min using a mini-volume air sampler (TAS, Airmetrics, Springfield, OR, USA) equipped with a two-stage particle size separation device. Particles less than 2.5 μm were collected on the filter, the coating film on the surface of the first impactor separated particles larger than 10 μm, and those larger than 2.5 μm were separated by the second impactor. We applied a thick coat of suspending solution prepared by dissolving grease in 30 mL of hexane to the impactor surface for efficient attachment of particles. A Zeflour^TM^ Teflon filter (Pall Corp., Port Washington, NY, USA, pore size: 2.0 μm, diameter: 47 mm) was used for sampling.

### 2.3. Determination of PM_2.5_ and Trace Element Concentrations

We determined the PM_2.5_ concentration by weighing the filter using an electronic balance (Sartorius M2P, Goettingen, Germany) capable of measuring to 0.001 mg after conditioning the filter in a desiccator for 24 h. We repeated the weighing at least three times and used the mean values. Filters with the collected samples were stored at −20 °C to prevent volatilization.

We analyzed 18 trace elements (As, Ba, Ag, Sr, V, Cr, Mn, Fe, Co, Ni, Cu, Zn, Pb, Cd, Si, Al, Se, and Ti) according to the United States Environmental Protection Agency (US EPA) Compendium Method IO−3.5 [38]. 18 trace elements were carefully selected based on previous studies which identified various sources using those markers in PM_2.5_ [26,27,28]. A Teflon filter was placed in a Teflon vessel with a diameter of 3.5 cm, and the cap was closed after adding 15 mL of 5% HNO_3_. The vessel was placed in a microwave oven (MARS Xpress, CEM, USA), and the temperature was increased from 25 to 200 °C at a rate of 10 °C/min, and maintained for 20 min. The eluate was filtered, transferred to a 15 mL tube, and analyzed using inductively coupled plasma-mass spectrometry (NexION 300D, PerkinElmer, Waltham, MA, USA). The analysis gas was 99.99% pure Ar, and the elements were detected with a quadrupole ion deflector equipped with a triple cone interface. The dynamic reaction cell (DRC-e) function was used for some elements, where ammonia gas was injected into the spray chamber before passing through the plasma to reduce the interference effect of elements with similar masses. Multi-Element Calibration Standard 3 and 5 (PerkinElmer, Waltham, MA, USA) were used for standard solutions. The following are the method detection limits for each element (μg/L): As (0.006), Ba (0.054), Ag (0.004), Sr (0.020), V (0.001), Cr (0.026), Mn (0.002), Fe (0.225), Co (0.005), Ni (0.013), Cu (0.010), Zn (0.311), Pb (0.001), Cd (0.003), Si (2.331), Al (0.860), Se (0.035), and Ti (1.172).

### 2.4. Data Analysis

PM_2.5_ is largely composed of inorganic ions, carbonaceous materials, and trace elements. While most inorganic ions are secondarily produced and carbonaceous materials originate from combustion sources, trace elements are generated from a variety of natural and anthropogenic sources [39,40,41]. Thus, they have been used to estimate a wide range of PM_2.5_ sources, which is advantageous for investigating the effects of harmful sources, as a significant portion of PM_2.5_ risk is caused by trace elements [42,43,44]. In this study, we conducted a varimax rotated factor analysis for the sum of trace element concentrations, which was regarded as a surrogate of the PM_2.5_ concentration, using SPSS for Windows 20.0 (IBM, Armonk, NY, USA). We analyzed 41 samples for 18 elements, which exceeds the minimum number of samples (*n*) suggested by Henry et al. as follows: *n* > 30 + (V + 3)/2 = 40.5, where V is the number of variables (elements in this study) [45].

We used the enrichment factor (EF) to distinguish anthropogenic influence from crustal origin [42,43,46,47]. Using Si, a representative element of the Earth’s crust, we calculated the EF using the relative ratio of an individual trace element to Si in PM_2.5_ to that in the Earth’s crust as follows:EF = (X/Si)_PM_/(X/Si)_crust_
where X and Si denote the concentrations of the trace element and Si, respectively, and the subscripts PM and crust denote PM_2.5_ and the Earth’s crust, respectively [2,39]. The values provided in Taylor were used for determining the trace element concentrations in the Earth’s crust [48]. If EF is close to 1, the element is considered to be of crustal origin. EF becomes greater than 1 when X is greater than the crustal origin concentration because of anthropogenic influence. Here, we assumed that Si was not emitted from anthropogenic sources.

As trace elements contained in PM may cause cancer when introduced into the human body through respiration, ECR was assessed using the following equation:ECR = X_ecr_ × (toxicity value)
where X_ecr_ denotes the 95th percentile concentration of the element [49]. The inhalation value provided by the Integrated Risk Information System (IRIS) [50] was used for the toxicity value. For Cr, ECR was calculated using the Cr(VI) concentration, which exhibits a high risk among all Cr concentrations, as Cr/7 because Cr(III) and Cr(VI) typically occur in a 6:1 ratio [51]. For Pb, we used the value provided by the Office of Environmental Health Hazard Assessment, California Environmental Protection Agency, because no quantitative risk was presented by IRIS [52].

## 3. Results and Discussion

### 3.1. PM_2.5_ and Trace Element Concentrations in Indoor and Outdoor Air

Table 1 shows the mean concentrations of PM_2.5_ and trace elements in indoor and outdoor air measured at the 41 USDs. The mean indoor PM_2.5_ concentration is 18.0 μg/m^3^, which is lower than the 24-h average standard mandated by the Indoor Air Quality Control Act (50 μg/m^3^). Furthermore, all the indoor PM_2.5_ concentrations measured at the 41 USDs complied with the standard. However, the mean outdoor PM_2.5_ concentration is 25.2 μg/m^3^, but approximately 20% (8 of 41 USDs) was exceeded the 24-h average standard for outdoor air (35 μg/m^3^ at the time of measurement). This shows that indoor PM_2.5_ concentrations in the USDs met the standard even with outdoor air quality being exceeded on multiple days. An indoor air quality study of USDs in Korea also reported that PM concentrations typically met the standard because indoor air quality was managed with air-conditioning using pre- and medium filters that facilitated the control of PM [53]. Yu et al. suggested that it is essential to provide air circulation (i.e., replace indoor polluted air with fresh air) in underground buildings because the indoor air is of lower quality compared to the indoor air quality of buildings at ground level [54].

The indoor-to-outdoor (I/O) ratio for PM_2.5_ concentration was calculated to be 0.76. Hu and Li measured PM_2.5_ at various points in shopping malls and reported that the I/O ratio ranged from 0.46 to 0.52, which is less than the values in Table 1 [55]. However, they observed outdoor PM concentrations of approximately 300 μg/m^3^, which is significantly greater than the concentrations observed in this study. However, Klinmalee et al. reported a PM_2.5_ I/O ratio of 1.5 for a department store, indicating a strong indoor source associated with the crowded conditions, despite a busy traffic outside [56]. Jones et al. measured PM concentrations in houses located close to roads and found that the I/O ratio was 1.0 ± 1.3, a value comparable to our results [57].

Trace elements accounted for 13.6% of indoor PM_2.5_ and 10.8% of outdoor PM_2.5_ concentrations. Ti was detected in the highest concentrations in both indoor and outdoor air, followed by Fe, Si, Zn, and Al. Ti accounted for approximately 50% of the total trace element concentrations. The sum of the top five elements (Ti, Fe, Si, Zn, and Al) was found to be approximately 90% of the total trace element amount. The sum of As, Cd, Cr, Ni, and Pb, all of which present high cancer risks, was 0.25% and 0.20% of the indoor and outdoor PM_2.5_ concentrations, respectively. These values are greater than those reported in the United States but less than those measured in Europe [34,35,58].

### 3.2. Trace Element Sources

The results from a varimax rotated factor analysis are summarized in Table 2. We identified five factors with eigenvalues greater than 1, explaining 83.8% of the total variance. Prior to specifying the source for each factor, we examined the characteristics of each factor using the correlation between indoor and outdoor concentrations, I/O ratio, and EF in Table 3. We used the concentration sum of the marker elements with high loading in Table 2 for the concentration of each factor by assuming that the marker elements only exist in that factor. For outdoor concentrations, we used the sum of all 18 element concentrations to calculate the correlation coefficient, while we used the sum of the marker element concentrations to calculate the I/O ratio. We used geometric means to calculate the EFs listed in Table 1 for factors because element EFs differ by orders of magnitude.

In Table 3, factor 1 has the lowest correlation with outdoor air (albeit a higher *p*-value than that of a statistical significance), and the I/O ratio is greater than 1. This implies that the indoor concentrations were higher than the outdoor concentrations, and mainly varied with the influences of indoor sources. Additionally, EF is the second highest after factor 4, indicating that the associated elements were largely enriched by anthropogenic sources. In addition to outdoor air, factors 2 and 4 exhibit lower correlations with factors 1, which have the first and second highest correlations with the outdoor air, respectively. The highest correlation of factor 2 with the outdoor air was because the variations in the element concentrations of factor 2 were closely related to those of the outdoor air. However, the lowest I/O ratio and highest EF for factor 4 suggests that the outdoor concentrations became higher than the indoor concentrations mainly due to anthropogenic influences. For factors 2 and 5, EFs are low and the I/O ratios are close to 1. Elements associated with these factors were mostly of crustal origin, and were ubiquitous in both indoor and outdoor air.

Factor 1 has high loadings for Cu, Cr, Se, Ni, Mn, and Ag, and accounts for 28% of the total variance. Considering the higher I/O ratios for elements such as Ag and Ni, and larger EF values for elements such as Se and Ag (Table 1), strong effects of anthropogenic indoor sources shown by factor 1 in Table 3 are plausible. As the number of marker elements is large, various sources can be considered. For example, we can consider soil dust as a source of Mn, and coal combustion for Se [2,59,60,61]. According to an extensive review conducted by Chow, Cr, Ni, and Mn are related to vehicle exhaust, and Cu and Ag are released from incinerators [39]. However, these sources are not specific to outdoor settings. Since all elements are metals, they are commonly used in the manufacturing of electric devices, cooking appliances, plastics, and jewelry, and used for pigments, painting, electroplating, and cosmetics [25,28,32]. Considering the nature of USDs, these elements appear to be associated with stores that stock and sell aforementioned types of manufactured products. These elements, except for Ag, are also found in cigarette smoke [62,63,64]. Cigarette smoke is likely a major source because its effects are more significant indoors than outdoors.

Factor 2 is heavily loaded with Sr, Si, and Ti, and exhibits 19% of the total variance. Ti and Si are representative crust constituents along with Al, Ca, Fe, and Mn [2,39,48,65,66]. In East Asia, concentrations of these elements are high during the Asian dust period [67,68,69]. In Table 1, Sr has the lowest EF except for Si and Al, indicating a representative crustal element. Despite contributing a small fraction, Sr is found in bare land, roadways, agricultural fields, construction sites, and deteriorated building materials [27,39,42,70]. In Table 3, the I/O ratio for factor 2 is 0.96, showing minimal differences between indoor and outdoor concentrations. We assume a significant amount of fugitive dust indoors as well as outdoors due to the large number of passers-by.

Factor 3 is responsible for 14% of the total variance, and is heavily loaded with Co, Zn, and V. Zn is widely used as a marker for vehicle exhaust and V originates from oil combustion [39,71,72,73,74]. Co is also associated with residual oil and fossil fuel combustion [39,75,76]. Because oil is not used as fuel in USDs, oil combustion as well as vehicle exhaust is not considered as an indoor source. However, Table 3 reveals that factor 3 has the second lowest correlation with outdoor air following factor 1. Among all factors, albeit lower than that with factor 4, factor 3 is highly correlated with factor 1, which shows a strong indoor influence. The fractions of Co, Zn, and V in gas combustion emissions were higher than those in other combustion emissions; therefore, the gas combustion can be considered a source of these elements [40,77]. These elements are presumed to be emitted from metals in the boiler and its associated ducting rather than from the fuel itself. Zn is also reported to be released during cooking with gas stoves [78,79,80,81]. However, in Table 3, unlike factor 1, the I/O ratio for factor 3, which is slightly lower than 1, indicates that the effects of the outdoor sources are larger than those of the indoor sources.

Factor 4 is responsible for 12% of the total variance. It shows high loadings for As, Pb, and Cd. As mentioned earlier, factor 4 represents anthropogenic outdoor sources, whereas factor 1 represents anthropogenic indoor sources. In addition, considering a high correlation with factor 3, we can consider a combustion-related source for factor 4. Since V and Ni, which are related to oil combustion [2,39,82,83] are excluded, factor 4 is likely to be associated with coal combustion [60,70,71,84,85]. In Korea, As, Pb, and Se are used as tracers of coal combustion emissions [61,86,87], and are presumed to result from long-range transport from China where a large amount of coal is consumed [88,89,90,91]. This suggests why factor 4 is the least correlated with factor 1 in Table 3. It is worth noting that all the elements in factor 4 have high carcinogenic risk and were used to calculate the ECR in this study.

Finally, factor 5 shows high loadings for Ba, Fe, and Al and accounts for 11% of the total variance. In Table 1, Fe and Al are classified as crustal elements because of their low EFs. Thus, the characteristics of factor 5 are similar to those of factor 2, but the correlation with the outdoor air is less than that for factor 2. Ba is used in brake pads, tire wear, and lubrication oil and is emitted as vehicle exhaust; hence, it is often found in road dust [11,92,93,94]. Fe is also found in road dust because it is a crustal element; however, Fe is especially high in subways because it is generated by friction and wear in the braking and supply of electricity to subway trains [95,96,97,98]. Despite the high proportion of crust elements, the correlation of factor 5 with outdoor air is lower, presumably because of dust generated in the subway system.

### 3.3. Contribution of Trace Element Sources by USD Environmental Factor

Table 4 shows PM_2.5_ concentrations and contributions of trace element sources according to the USD environmental factor. Because the PM_2.5_ I/O ratio in Table 1 was 0.76 (overall ratio in Table 4), most I/O ratios for PM_2.5_ are less than 1.0 regardless of the environmental factor. The I/O ratio only approaches 1 when the USD entrance is open and ventilation is prevalent (Table 4c). As the entrance changes to semi-open and closed, the I/O ratio decreases to 0.82 and 0.68, respectively. Overall, the contribution of soil dust is the largest at 67%, that of road/subway dust is 24%, with a combined contribution of approximately 90% (Table 4a). However, coal combustion and miscellanea, which contain harmful elements, contribute only 0.7% and 3.4%, respectively. The contribution of coal combustion composed of elements with high carcinogenic risk is highest in Seoul, followed by Incheon and then Gyeonggi. However, the concentration of coal combustion is the highest in Incheon because the total element concentration is high (Table 4b). Incheon experiences significant amounts of dust because of its proximity to the port and China, and because a large coal-fired power plant is located approximately 30 km southwest [99,100]. This causes high concentrations of coal combustion elements even indoors.

The element concentration in Incheon due to indoor miscellanea is twice that of Seoul, but with similar contributions of 3.7% and 3.3% as the element concentration mainly varies with the total concentration. The concentration due to miscellanea increases with the number of stores, but the greatest contribution occurs when the number of stores is 100–199 because the increase in total concentration is larger (Table 4e). The effect of the number of stores is most distinct in soil dust, which accounts for a large portion of the total concentration. As the number of stores increases, both absolute concentration and contribution increase, which is likely due to the increased number of passers-by. The effect on the contribution is most pronounced in the type of stores (Table 4d). As the type of stores changes from open to mix and closed, the contribution of most sources decreases, while soil dust contribution increases. If the stores are closed, the passage area volume in the USD is reduced, which causes the same amount of generated soil dust to have an increased contribution. When the stores are closed, the contribution of miscellanea decreases, but the concentration increases. This phenomenon can result either from the decrease in the volume of the passage area as in soil dust, or from emissions occurring mainly outside the store and not inside. The emissions from miscellanea occur during stocking and handling of various goods. Another significant source of miscellanea includes smoking. If smoking mainly occurs outside the store, the concentration may increase when the stores are closed. However, even taking this into account, the effect of volume reduction of the passage area is presumed to have a greater impact. We confirm this by observing decreases in vehicle exhaust/cooking and road/subway dust contributions, comprising both indoor and outdoor emissions, as well as in miscellanea.

The effects of indoor and outdoor emissions are more easily distinguished by examining the variation based on USD exit type (Table 4c). In contrast to miscellanea and soil dust, whose variations are unclear, the contributions of vehicle exhaust/cooking and coal combustion increase and decrease, respectively, when the USD exit is closed. This demonstrates that indoor emissions are more impacted by vehicles exhaust/cooking, whereas outdoor emissions are impacted by coal combustion. The effect of road/subway dust is most evident in Table 4f when comparing if a USD is connected to the subway. When connected to the subway, the concentrations by most sources except miscellanea decrease because of the increase in the volume of the passage area, but the concentration of road/subway dust increases by 1.7 times and the contribution increases by more than 10%.

### 3.4. Excess Cancer Risk (ECR)

We calculated the ECR through respiration using the 95th percentile concentrations and toxicity values for five elements with carcinogenic risk, as shown in Table 5. The carcinogenic risk caused by inhaling the elements was greatest in Cr followed by As, Cd, Ni, and Pb. We suggest in Section 3.2 that As, Cd, and Pb result from coal combustion, and Cr and Ni result from indoor miscellanea. The concentrations of Cr and Ni from indoor miscellanea are high indoors, whereas those of As and Cd from coal combustion are high outdoors. Pb has an outdoor air quality standard of 5 μg/m^3^ per year in Korea. Despite using the 95th percentile, the concentration in Table 5 is 0.7% of the standard.

The ECR is 0.3–67 indoors and 0.4–60 outdoors per million people. The US EPA sets a goal for ECR of 10^−6^ and 10^−4^ based on the maximum concentration near the pollutant source (1 and 100 persons per million, respectively) [101]. However, Table 5 shows that the ECR for Pb is less than 10^−6^, and even the highest ECR for Cr is less than 10^−4^. In Korea, ECRs were investigated only for the outdoors. ECRs for Cr, As, and Cd in Seoul were 25–54 per million people [102], which are comparable to those listed in Table 5. ECRs for Cd, Cr, and Ni in Ulsan were 8.4–35 per million people, which do not exceed those listed in Table 5, even though the study area is industrial [51].

## 4. Summary and Conclusions

We investigated the characteristics of PM_2.5_ and trace elements in underground shopping districts (USDs) located in the Seoul metropolitan area. We estimated their sources and assessed the cancer risk from respiratory exposure of these trace elements for workers and users in USDs.

PM_2.5_ concentrations in 41 USDs did not exceed the standard established by the Indoor Air Quality Control Act. The PM_2.5_ I/O ratio was 0.76, indicating that the indoor concentration was lower than the outdoor value. Among the PM_2.5_ trace elements, Ti, a substance originating from the Earth’s crust, exhibited the highest concentration, followed by Fe, Si, and Zn. Using a varimax rotated factor analysis, we identified five sources for the elements: indoor miscellanea, soil dust, vehicle exhaust/cooking, coal combustion, and road/subway dust. The overall contribution of miscellanea, which has a strong effect on indoor sources, was 3%, whereas that of soil dust, which is of crustal origin, was 67%. Vehicle exhaust/cooking and road/subway dust are composed of both outdoor (vehicle exhaust, road dust) and indoor (cooking, subway dust) sources. Higher contribution of vehicle exhaust/cooking when the USD entrance was closed indicated a larger effect of indoor emissions. However, we determined that road/subway dust concentration and contribution were greater when the USD was connected to the subway. The contribution of coal combustion is higher in Incheon, which is near the border with China as well as large-scale coal-fired power plants. Coal combustion contributions increased when the USD entrance was changed from closed to semi-open and open. Both implied that the effects of outdoor emissions are larger for coal combustion. The effects of indoor and outdoor emissions were almost the same for soil dust, but the increase in the contribution to the number of stores was evident, presumably due to the increase in fugitive emissions with an increase in the number of passers-by.

The influence of anthropogenic emissions that were investigated using the enrichment factor was higher for miscellanea and coal combustion. Among the elements having carcinogenic risk, Cr and Ni were included in miscellanea, and Pb, Cd, and As were included in coal combustion. The excess cancer risk (ECR) using the 95th percentile concentration was the highest at 67 × 10^−6^ for Cr, but less than 10^−4^, and the ECR for Pb was lower than 10^−6^. In Korea, ECRs were estimated outdoors in Seoul and Ulsan (an industrial area), which were comparable to and support the study results.

## Figures and Tables

**Figure 1 ijerph-18-00297-f001:**
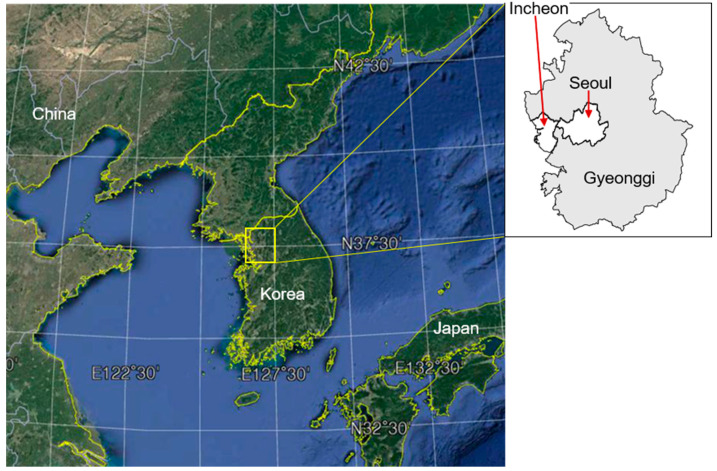
Locations of Seoul metropolitan area (SMA; yellow box in left figure), including Seoul proper, Incheon, and Gyeonggi (right panel). The background map was obtained from © Google Maps.

**Table 1 ijerph-18-00297-t001:** Concentrations of PM_2.5_ (μg/m^3^) and trace elements (ng/m^3^) in indoor and outdoor air (mean ± standard deviation).

	Indoor	Outdoor	I/O Ratio ^a^	R ^b^	EF ^c^
PM_2.5_	18.0 ± 8.0	25.2 ± 10.6	0.76 ± 0.31	0.72 **	-
Ag	0.337 ± 0.473	0.337 ± 0.717	1.41 ± 1.48	0.08	2170
Ni	9.22 ± 8.26	9.46 ± 7.17	1.34 ± 1.16	0.16	78
Ti	1240 ± 1380	1380 ± 2020	1.16 ± 0.65	0.38 *	134
Al	73.2 ± 64.1	96.1 ± 84.6	1.15 ± 1.18	0.48 **	1
Zn	129 ± 85	123 ± 60	1.11 ± 0.68	0.51 **	1248
Co	1.61 ± 1.71	1.46 ± 1.04	1.07 ± 0.39	0.78 **	40
Fe	486 ± 376	482 ± 263	1.07 ± 0.82	0.45 **	6
Cr	18.5 ± 23.3	15.8 ± 11.8	1.03 ± 0.32	0.31	107
Cd	2.29 ± 1.12	2.65 ± 1.96	1.01 ± 0.42	0.63 **	8178
Mn	20.3 ± 12.4	22.3 ± 11.4	0.95 ± 0.44	0.68 **	15
Sr	2.64 ± 1.93	3.10 ± 2.78	0.93 ± 0.27	0.86 **	5
Cu	21.2 ± 18.9	25.9 ± 19.2	0.90 ± 0.59	0.22	259
Se	14.0 ± 12.3	17.1 ± 13.0	0.89 ± 0.49	0.31 *	166,960
Si	379 ± 179	481 ± 291	0.87 ± 0.31	0.78 **	1
Ba	17.2 ± 15.6	21.9 ± 10.1	0.81 ± 0.65	0.34 *	26
V	5.87 ± 6.55	8.23 ± 9.11	0.75 ± 0.21	0.86 **	21
As	1.98 ± 1.26	2.96 ± 2.02	0.69 ± 0.21	0.89 **	721
Pb	13.5 ± 6.7	20.1 ± 9.3	0.69 ± 0.19	0.80 **	771

^a^ Indoor/outdoor ratio. Trace elements are arranged in descending order of I/O ratio. ^b^ Correlation coefficient, *p*-value: ** *p* < 0.01, * *p* < 0.05. ^c^ Geometric mean of enrichment factor for indoor elements.

**Table 2 ijerph-18-00297-t002:** Factor loadings from principal component analysis with varimax rotation. Boldface denotes high factor loadings considered as marker elements.

	Factor 1	Factor 2	Factor 3	Factor 4	Factor 5
Cu	**0.956**	0.078	0.115	0.086	0.135
Cr ^a^	**0.938**	0.117	0.099	0.139	0.073
Se	**0.893**	0.102	0.323	0.095	0.149
Ni ^a^	**0.835**	−0.021	0.231	−0.066	−0.040
Mn	**0.740**	0.143	0.347	0.308	0.043
Ag	**0.716**	0.332	−0.247	−0.088	−0.036
Ti	0.133	**0.944**	0.204	0.071	0.072
Sr	0.105	**0.927**	0.102	0.195	0.211
Si	0.193	**0.808**	0.001	0.393	0.236
Co	0.236	0.050	**0.859**	−0.107	0.154
Zn	0.325	0.259	**0.688**	0.104	0.151
V	0.042	0.020	**0.683**	0.258	−0.244
As ^a^	−0.065	0.120	0.146	**0.883**	0.122
Cd ^a^	0.261	0.290	−0.018	**0.768**	0.079
Pb ^a^	0.121	0.451	0.534	**0.581**	−0.101
Ba	−0.025	0.116	−0.075	0.056	**0.948**
Al	0.191	0.542	0.114	0.212	**0.667**
Fe	0.611	0.295	0.151	0.057	**0.645**
Eigenvalue	7.55	3.02	2.08	1.33	1.12
% variance	28.2	18.5	13.6	12.2	11.4
Cumulative % variance	28.2	46.7	60.3	72.4	83.8
Possible source	Indoor miscellanea	Soil dust	Vehicle exhaust/cooking	Coal combustion	Road/subway dust

^a^ Of high carcinogenicity; hence, used to assess the excess cancer risk.

**Table 3 ijerph-18-00297-t003:** Correlations between factors and outdoor air ^a^, mean indoor-to-outdoor (I/O) ratios, and enrichment factors (EFs).

	Factor 1	Factor 2	Factor 3	Factor 4	Factor 5
Factor 1	1				
Factor 2	0.30	1			
Factor 3	0.49 **	0.42 **	1		
Factor 4	0.34 *	0.56 **	0.52 **	1	
Factor 5	0.66 **	0.50 **	0.43 **	0.32 *	1
Outdoor	0.26	0.86 **	0.31 *	0.60 **	0.41 **
I/O ratio ^b^	1.10	0.99	0.98	0.80	1.01
EF ^c^	470	9	103	1656	4

^a^ Used the sum of the marker element concentrations for each factor and the total of all 18 element concentrations for the outdoor air. *p*-value: ** *p* < 0.01, * *p* < 0.05. ^b^ Used the sum of the marker element concentrations for outdoor air as well. ^c^ For indoor elements.

**Table 4 ijerph-18-00297-t004:** PM_2.5_ concentrations and contributions of trace element sources ^a^ by underground shopping district (USD) environmental factor.

		Number of Data Points	PM_2.5_ (μg/m^3^)				Element Concentration (ng/m^3^) ^b^						Contribution (%) ^c^			
			Indoor	Outdoor	I/O ratio	Total	Indoor Miscel.	Soil Dust	Vex/ Cook.	Coal Comb.	Rd/sw dust	Indoor Miscel.	Soil Dust	Vex/ Cook.	Coal Comb.	Rd/sw Dust
(a) Overall		41	18.0	25.2	0.76	2439	83	1625	136	18	576	3.42	66.6	5.59	0.73	23.6
(b) Location																
	Seoul	24	17.9	25.1	0.72	1964	64	1278	101	14	508	3.25	65.0	5.15	0.69	25.9
	Incheon	15	18.2	25.8	0.81	3363	123	2286	205	25	723	3.66	68.0	6.10	0.75	21.5
	Gyeonggi	2	16.5	21.8	0.75	1200	22	841	44	12	281	1.79	70.1	3.67	1.02	23.4
(c) Type of USD																
	Open	3	30.2	31.2	1.00	2037	69	1316	102	20	531	3.40	64.6	4.98	0.97	26.1
	Semi-open	15	17.0	22.5	0.82	2771	106	1904	143	23	595	3.83	68.7	5.14	0.83	21.5
	Closed	23	17.0	26.1	0.68	2274	71	1483	137	14	570	3.10	65.2	6.01	0.62	25.0
(d) Type of stores																
	Open	23	17.0	24.4	0.77	2461	97	1537	152	19	656	3.93	62.5	6.16	0.78	26.7
	Mix	3	20.3	27.6	0.72	1868	58	1315	100	15	379	3.12	70.4	5.36	0.81	20.3
	Closed	15	14.0	19.0	0.87	5126	107	3851	200	20	949	2.08	75.1	3.89	0.38	18.5
(e) Number of stores																
	<99	17	20.5	28.1	0.75	1781	58	1116	101	14	492	3.28	62.6	5.64	0.78	27.7
	100–199	11	14.7	24.9	0.60	1851	68	1246	128	18	391	3.68	67.3	6.89	0.97	21.1
	>200	13	17.5	21.7	0.90	3796	129	2612	191	23	842	3.40	68.8	5.02	0.60	22.2
(f) Connection to subway																
	Yes	23	17.1	23.6	0.78	2465	89	1522	131	17	707	3.61	61.7	5.30	0.67	28.7
	No	18	19.0	27.2	0.72	2405	77	1757	143	19	409	3.18	73.1	5.96	0.80	17.0

^a^ Indoor miscel., indoor miscellanea; Vex/cook., vehicle exhaust/cooking; Coal comb., coal combustion; Rd/sw dust, road/subway dust. ^b^ Sum of the marker element concentrations for each source. ^c^ Concentration by source divided by the total element concentration.

**Table 5 ijerph-18-00297-t005:** Excess cancer risks (ECR) of carcinogenic elements.

	IARC Classification Group	Tumor Type	Toxicity Value (m^3^/μg)	Concentration (ng/m^3^) ^a^		ECR (10^−6^)	
				Indoor	Outdoor	Indoor	Outdoor
Cr(IV) ^b^	A	Lung	1.2 × 10^−2^	5.6	5.0	66.6	59.9
As	A	Lung	4.3 × 10^−3^	4.3	5.4	18.5	23.2
Cd	B1	Lung, trachea, and bronchus	1.8 × 10^−3^	4.3	6.8	7.8	12.2
Ni	A	Lung	2.4 × 10^−4^	23.0	17.2	5.5	4.1
Pb	B2	Lung	1.2 × 10^−5^	25.1	34.7	0.3	0.4

^a^ 95th percentile. ^b^ Cr/7.

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
