# Peer review of "PM2.5 and Trace Elements in Underground Shopping Districts in the Seoul Metropolitan Area, Korea"

_ijerph, 2021, doi:10.3390/ijerph18010297_

Round 1
Reviewer 1 Report
In this paper, authors present the result of the measuramenmts of PM2.5 in 41 underground shopping districts in the Seoul metropolitan area from June to November 2017, and associated trace elements to determine the 13 sources and assess the respiratory risks. The introduction provide sufficient information and the bibliography is sufficiently up to date. The applied methodolog is well described and the ststistical results are analized to determine the origins of the pollution. Results area commented and critically analyzed, too.
I suggest only some improvement in the presentation of the data: for example, tables 1 and 2 could result more readible if the reported values are integrated with some figure (e.g. bar charts)
Author Response
Response to Reviewer #1
General comments:
In this paper, authors present the result of the measurements of PM2.5 in 41 underground shopping districts in the Seoul metropolitan area from June to November 2017, and associated trace elements to determine the 13 sources and assess the respiratory risks. The introduction provided sufficient information and the bibliography is sufficiently up to date. The applied methodology is well described and the ststistical results are analized to determine the origins of the pollution. Results area commented and critically analyzed, too.
Response: We thank the reviewer for carefully reviewing the manuscript and providing valuable comments. The following are our responses to your specific comments.
- I suggest only some improvement in the presentation of the data: for example, tables 1 and 2 could result more readible if the reported values are integrated with some figure (e.g. bar charts). 

We agreed with reviewer’s suggestion for improving the readability by replacing tables to figures. However, when we considered 18 trace elements or 19 components (with PM2.5), tabulating the mean and standard deviation (Table 1) and factor loadings from PCA (Table 2) is the best choice to avoid complexity of figure and to easily cite our results for future studies. Therefore, we decided to leave the tables.
Reviewer 2 Report
The paper "PM2.5 and Trace Elements in Underground Shopping Districts in the Seoul Metropolitan Area, Korea" shows the sources and the respiratory risks of trace elements concentrations in PM2.5. The measurements were carried out in 41 underground shopping districts and outdoor in the Seoul .metropolitan area from June to November 2017
I think that the paper can be accepted for publication after a few minor revisions according to my following comments.
Comment 1. Line 91-92: Scale is missing (Fig. 1).
Comment 2: 115-116 The Authors are invited to explain their criterion for the choice of elements.
Comment 3: What certified reference materials were used?
Comment 4: Line 165-166 The sentence needs to be clarified - standard for outdoor air (35 μg/m3) is exceeded by approximately 20%? Please clarify.
Author Response
Response to Reviewer #2
Summary:
The paper "PM2.5 and Trace Elements in Underground Shopping Districts in the Seoul Metropolitan Area, Korea" shows the sources and the respiratory risks of trace elements concentrations in PM2.5 The measurements were carried out in 41 underground shopping districts and outdoor in the Seoul metropolitan area from June to November 2017.
I think that the paper can be accepted for publication after a few minor revisions according to my following comments.
We thank the reviewer for reviewing the manuscript. Your comments are italicized for convenience. The line (L) numbers in the responses correspond to those in the revised manuscript. The changes in the revised manuscript are underlined in the responses as necessary, and important changes are highlighted in yellow in the highlighted version of the revised manuscript.
- L91-92: Scale is missing (Fig. 1).
We have added longtitude and latitude in Figure 1, instead of scale as follows:
Figure 1. Locations of Seoul metropolitan area (SMA; yellow box in left figure), including Seoul proper, Incheon, and Gyeonggi (right panel). The background map was obtained from © Google Maps.
- L115-116 The Authors are invited to explain their criterion for the choice of elements.
The reason for selecting 18 trace elements is that those elements are the key species for identifying natural and anthropogenic sources, such as soil dust, vehicle exhaust fumes, coal combustion, subway, etc. Thus, we added description of reason for selecting 18 trace elements as follows:
“18 trace elements were carefully selected based on previous studies which identified various sources using those markers in PM2.5 [26-28].” (L118-119).
- What certified reference materials were used?
We added specific information of standard solutions as follows:
“Multi-Element Calibration Standard 3 and 5 (Perkin Elmer, USA) were used for standard solutions.” (L129-130).
- L165-166 The sentence needs to be clarified-standard for outdoor air (35 μg/m3) is exceeded by approximately 20%? Please clarify.
We clarified the relevant sentence as follows:
“However, the mean outdoor PM2.5 concentration is 25.2 μg/m3, but approximately 20% (8 of 41 USDs) was exceeded the 24-h average standard for outdoor air (35 μg/m3 at the time of measurement) is exceeded by approximately 20%.” (L168-170)